A tragedy of the commons case study: modeling the fishers king crab system in Southern Chile

Zambrano Alan alanrzh@fundacionbariloche.org.ar alanrzh@gmail.com 1
Laguna María F. 2
Kuperman Marcelo N. 2 3
Laterra Pedro 1
Monjeau Jorge A. 1
Nahuelhual Laura 4 5 6
1 Fundación Bariloche and CONICET , San Carlos de Bariloche , Argentina
2 Centro Atómico Bariloche - CONICET, Comisión Nacional de Energía Atómica , San Carlos de Bariloche , Río Negro , Argentina
3 Instituto Balseiro, Universidad Nacional de Cuyo , Cuyo , Mendoza , Argentina
4 Departamento de Ciencias Sociales, Universidad de Los Lagos , Osorno , Chile
5 Instituto Milenio en Socio-Ecología Costera , Santiago , Chile
6 Centro de Investigación, Dinámica de Ecosistemas Marinos de Altas Latitudes , Chile
Hedrick Brandon
Electronic publication date: 2023 Mar 14
Publication date: 2023
Volume: 11
Electronic Location ID: e14906
Received 2021 Jul 12; Accepted 2023 Jan 25
Copyright: ©2023 Zambrano et al.
Copyright year: 2023
Copyright holder: Zambrano et al.
License: This is an open access article distributed under the terms of the Creative Commons Attribution License, which permits unrestricted use, distribution, reproduction and adaptation in any medium and for any purpose provided that it is properly attributed. For attribution, the original author(s), title, publication source (PeerJ) and either DOI or URL of the article must be cited.
License URL: https://creativecommons.org/licenses/by/4.0/

Keywords: Networks, Game theory, Allee-Efect, Time-dependent dynamical system, Artisanal fisheries, Illegal fishing, Centolla

Funding: “Fondo para la Investigación Científica y Tecnológica of Argentina” (FONCYT, PICT 2015 0672) The Inter-American Institute for Global Change Research (IAI) CRN3 095 US National Science Foundation GEO-1128040 FONDAP 15150003 This study was funded by “Fondo para la Investigación Científica y Tecnológica of Argentina” (FONCYT, PICT 2015 0672), the Inter-American Institute for Global Change Research (IAI) CRN3 095, which is supported by the US National Science Foundation (Grant GEO-1128040), and FONDAP (Grant 15150003). The funders had no role in study design, data collection and analysis, decision to publish, or preparation of the manuscript.

==============================
Illegal fishing in small-scale fisheries is a contentious issue and resists a straightforward interpretation. Particularly, there is little knowledge regarding cooperative interactions between legal and illegal fishers and the potential effects on fisheries arising from these interactions. Taking the Chilean king crab (Lithodes santolla; common name centolla) fishery as a case study, our goal is twofold: (i) to model the effect of illegal-legal fishers’ interactions on the fishery and (ii) analyze how management and social behavior affect fishery’s outcomes. We framed the analysis of this problem within game theory combined with network theory to represent the architecture of competitive interactions. The fishers’ system was set to include registered (legal) fishers and unregistered (illegal) fishers. In the presence of unregistered fishers, legal fishers may decide to cooperate (ignoring the presence of illegal fishers) or defect, which involves becoming a “super fisher” and whitewashing the captures of illegal fishers for a gain. The utility of both players, standard fisher and super fisher depend on the strategy chosen by each of them, as well as on the presence of illegal fishers. The nodes of the network represent the legal fishers (both standard and super fishers) and the links between nodes indicate that these fishers compete for the resource, assumed to be finite and evenly distributed across space. The decision to change (or not) the adopted strategy is modeled considering that fishers are subjected to variable levels of temptation to whitewash the illegal capture and to social pressure to stop doing so. To represent the vital dynamics of the king crab, we propose a model that includes the Allee effect and a term accounting for the crab extraction. We found that the super fisher strategy leads to the decrease of the king crab population under a critical threshold as postulated in the tragedy of the commons hypothesis when there are: (i) high net extraction rates of the network composed of non-competing standard fishers, (ii) high values of the extent of the fishing season, and (iii) high density of illegal fishers. The results suggest that even in the presence of super fishers and illegal fishers, the choice of properly distributed fishing/closure cycles or setting an extraction limit per vessel can prevent the king crab population from falling below a critical threshold. This finding, although controversial, reflects the reality of this fishery that, for decades, has operated under a dynamic in which whitewashing and super fishers have become well established within the system.

Introduction

Faced with the exploitation of a common resource, is it convenient for an individual to be selfish or cooperative? Which of the two strategies is the one that maximizes everyone’s benefits in the long term? Which of the two strategies maximizes the sustainability of the species? This dilemma and its variants have ignited a controversy that has been sustained for several centuries and in which several disciplines are intertwined in a dialectical dynamic: Top-down versus bottom-up regulatory forces of the management of the system, individual versus society, restrictions versus rights, sustainability versus collapse, local versus global goals.

The tragedy of the commons (Hardin, 1968) spawned a famous controversy about the outcomes of laissez-faire in the management of the commons. Hardin argued that if everyone is selfish, competing among the actors to obtain the maximum individual benefit, the system collapses sooner or later. According to Berkes (1985), Hardin’s paradigm is a useful way of analyzing many cases of collapse of fishery resources. However, ‘tragedy’ is not a universal feature of all fisheries for a number of reasons, including the fact that individual interests are often subservient to the collective interests of a community (Feeny, Hanna & McEvoy, 1996; Leal, 1998; Basurto, 2005). Therefore, it is necessary to try to explain the existing cases of sustainable use of resources in terms of the violation of the assumptions that underlie the paradigm.

Illegal fishing and noncompliance with rules are acknowledged as significant threats to sustainability worldwide (Agnew et al., 2009; Bianchi et al., 2014) and do not only regard industrial fleets on high seas but also small-scale fisheries (henceforth, SSF) (Cavole, Arantes & Castello, 2015; Luomba, Chuenpagdee & Song, 2016; Cepić & Nunan, 2017; Satizábal et al., 2021). Like other conservation crimes (Von Essen et al., 2014), they are the two faces of a multifaceted and complex phenomenon that encompasses a variety of behaviors, motivations and justifications (Battista et al., 2018) and can undermine management strategies and promote conflicts between resource users and regulation agencies, ultimately affecting fisheries sustainability (Cepić & Nunan, 2017; Okeke-Ogbuafor, Gray & Stead, 2020). SSF are particularly exposed to the adverse effects of illegal fishing and noncompliance (Hauck, 2008; Battista et al., 2018; Nahuelhual et al., 2020).

In this work we have developed a model that aims to mathematically characterize this problem for the particular case of the king crab (L. santolla) fishery in southern Chile, which has been extensively studied (Nahuelhual et al., 2020) and it is related to a well-known history of the collapse of resources due to overuse in the case of the crab fishery in Alaska (Dew & McConnaughey, 2005a). Besides, it contains all the necessary elements to model the trade-offs between individuals and society that the tragedy of the commons highlights. Illegality appears as a great threat to sustainability in the same sense as the laissez-faire proposed by Hardin (1968), that is, individuals in uncontrolled competition for the appropriation of common resources prioritizing individual benefit, which leads to unsustainability, especially because illegality eludes any type of control over resources.

Research on this topic arises from two schools of thought (Cepić & Nunan, 2017). The first is the instrumental approach based on rationalist deterrence and law enforcement models, which assumes that rational actors calculate the costs and benefits of their actions (Becker, 1968). According to this logic, individuals will choose to comply (or not) with regulations based on economic gains, the probability of being caught, and the severity of sanctions (Sutinen & Kuperan, 1999; Hatcher et al., 2000; MacKeracher et al., 2021). The second school emerges in response to the growing evidence that compliance can be achieved even when formal law enforcement is weak, and it recognizes that norms and morals, as well as the legitimacy of law and governance, are important factors influencing fishers’ decisions (Gezelius, 2002; Sutinen & Kuperan, 1999; Jagers, Berlin & Jentoft, 2012; Battista et al., 2018). A common approach to modeling behavior and non-compliance of fishers, usually inscribed within the first school of thought, is game theory (for comprehensive revisions see Kaitala, 1986; Munro, 2009; Bailey, Sumaila & Lindroos, 2010; Hannesson, 2011; Grønbæk et al., 2020a; Grønbæk et al., 2020b; Grønbæk et al., 2020c; Grønbæk et al., 2020d). Most applications focus on fisheries at high seas (e.g., Byers & Noonburg, 2007; Hannesson, 2011; Da Rocha, Villasante & González, 2013; Jensen et al., 2015; Bellanger et al., 2019; Davis & Harasti, 2020), considering both cooperative and non-cooperative games. In some cases, aspects related to individual psychology and social pressure are appealed to account for fishers’ own convictions and the need to be accepted by their community (e.g., Sutinen & Kuperan, 1999; Hatcher et al., 2000; Dietz, Ostrom & Stern, 2003; Kraak, 2011).

One of the archetypal games to depict the confrontation between cooperation and defection is the Prisoner’s Dilemma game, with several applications to fisheries (e.g., Cole, Izmalkov & Sjöberg, 2014; Zhang, 2021). In this game, the players may choose between two different strategies: defect or cooperate. Without giving up the ambition of obtaining some personal gain, cooperative and defective strategies compete and spread among the population. The origin of the dilemma lies in the fact that although from the individual perspective the optimal rational strategy is the defective one, cooperation is always the best collective choice. However, unless some kind of coordination arises or, in the case of spatially extended games, grouping among the cooperators emerges, cooperation will not prevail, leading the population to adopt a behavior that may even be self-destructive. This is the rationale to adopt the scheme posed by the prisoner’s dilemma in this work.

The most interesting features of the emergence of cooperation in a competitive game manifest themselves when considering spatially distributed games, as is the case when the players are located on top of a network. This is the case in the present work, where we are also considering a multiplayer game, meaning that at each round many players will play and their payoff will be accordingly calculated.

The tragedy of the commons (Hardin, 1968) is usually taken to be an example of the Prisoner’s Dilemma, as it is a problem of collective action (López et al., 2005) though it is illustrative of failed cooperation scenarios.

In this study, we conceptualize the interactions between registered fishers as a multiplayer version of the classical Prisoner’s Dilemma game in the presence of unregistered fishers whose effects will be evaluated. While the Prisoner’s Dilemma game has been applied to fisheries and non-compliance before, we identified some gaps to which this work can contribute: few studies focus on noncompliance within SSF; these studies do not model interactions between illegal and legal fishers, but focus on non-compliant fishers; they generally assume that both illegal and legal fishers act as rational agents, excluding behavior outside of this logic (individual optimal choices); most of them do not include non-compliant fishers’ moral preferences.

It is worth recalling that, although in the Prisoner’s Dilemma game the optimal social strategy is cooperation, the equilibrium solution corresponds to non-cooperation. As an outcome of this situation, the emergence of free riders moved by individualistic interests can undermine cooperation agreements on fisheries and lead to complete non-cooperation and eventual over-exploitation of the resource; the advantage of cooperation should be understood as the development of a sustainable activity. In this regard, assuming that there are no interactions between fishers can lead to misinterpretations of the results, both technical and social. Interactions can occur for reasons other than self-interest and it is why our model has included parameters that depict moral preferences that can influence decision-making (Borda, 2011).

In this context we aim at answering the following questions:

• Which strategy (standard or super fisher) achieves the maximum individual benefit in the long term?

• How does the proper selection for the independent variables affect the king crab population in the long term?

• Are there conditions under which illegal fishing might not affect the king crab population?

Methods

Case study

King crab is mainly caught in the Magallanes region in southern Chile. It is a high-value fishery and most crab is exported to foreign markets. The export chain is the most important process in terms of trade magnitude and value, with at least eight interacting agents: vessel owners, crew, intermediaries, processing plants, exporters, importers, distributors, and consumers. According to the commercial strategy of the processing plants, after cooking and storing the crabs, the product can be sold to exporters or directly shipped to foreign markets. In recent years, exports have concentrated on the Chinese market, with 80.71% and 79.63% of the volumes exported in 2018 and 2019, respectively. From 2018 to 2020, prices fluctuated between 4 and 4.9 USD/kg at the beginning of the season, and between 6.71 and 14.45 USD/kg at the end of the season. The average annual processing volume in the last five years was approximately 3,851 tons. In 2019, there were 18 processing plants in operation, which varied in size and level of specialization, although most of them processed more than one species. A minor fraction of the catches is distributed through the processing plants in the Chilean market. In 2020, 584 vessels were operating in the region, representing 73% of the region’s fishing fleet. The authorized fleet comprises: (i) vessel owners, (ii) transporters and (iii) fishers of the crew. Of the total fleet, 84% corresponds to vessels less than 12 m in length (SERNAPESCA, 2020).

At present, the fishery operates under a semi-open access regime (no quotas), and it is managed through sex (only male specimens can be captured), size (> 120 mm of carapace length), and season (July 1st to November 30th) measures, fishing gear regulations (only iron traps, no nets), and fishers’ exclusion through the Artisanal Fishing Registry (RPA for its Spanish name: Registro Pesquero Artesanal) that grants vessels and fishers the authorization to extract a given species (Nahuelhual et al., 2018). Historically, the RPA for king crab in the region has been set under 600 vessels, and it will remain like that in the future management plan. Unregistered vessels (without RPA) might reach half this number according to estimations of the region’s management committee for king crab and snow crab (Paralomis granulosa) which comprises fishers, processors, researchers, NGOs, and government representatives (Nahuelhual et al., 2020).

In this study, we are concerned with the RPA violation that entitles a particular practice denominated whitewashing, which involves vessel owners and intermediaries and consists of the transfer at sea of crabs from vessels without RPA to fishing or carrier vessels with RPA which “whitewash” the undeclared catch. The latter is locally known as “super fishers” because their declared catches exceed their landing capacity (vessel size, crew size, and fishing gear). Whitewashing is sometimes justified on a moral and solidarity basis since the activity is mainly dominated by families and networks of close friends.

In synthesis, the king crab fishery is a suitable case for our purpose for at least the following reasons: (i) it is an SSF but with high extraction rates which are correlated with the high international prices paid for the product; (ii) it is among the very few fisheries managed without quotas, which would more closely reflect a common-pool resource; the absence of a quota means that each vessel captures as many crabs as possible without restriction, other than the vessel and crew capacity to handle traps and load; (iii) in the last years and for unknown reasons, it has exhibited decreasing landings despite the suspected increasing fishing effort (rising number of traps per fishing vessel) (Nahuelhual et al., 2018; Nahuelhual et al., 2020).

Setting up the game

To characterize the studied system, we model the interaction between the registered fishermen by using an undirected network, where the nodes represent the registered fishermen and the links the intensity of the competitive interaction mediated by the interference parameter, assumed fixed. The color of the nodes encodes the strategy, in the game theory sense, that the registered fishers decide to choose in a given time step in the presence of the unregistered fishers. And finally, the unregistered fishers are modeled by a middle field that permeates this network of fishers.

In the subsection “Interaction among fishers within the king crab fishery system” we define the intensity of competitive interactions between fishermen as well as the strategies they use. In the subsection “Fishing strategies and utilities” we explicitly define the fishing strategies for the multiplayer game as a function of the other parameters in the system.

Interactions among fishers within the king crab fishery system

To capture the spatial distribution of the vessels, we did not consider a mean-field model that would correspond to a situation where everybody interacts with the whole community of vessels. To mimic the architecture of interaction between standard and super fishers, we considered a network whose nodes represent all the registered vessels (i.e., with RPA). The network extends over the totality of fishing sites. The links between nodes represent the interaction among vessels competing for the local extraction of the king crab. We assumed that the geographical extent of each fishing site is large; consequently, the network is not very dense, meaning that the connections are diluted but not to the extreme of considering a system of isolated fishers. When two registered vessels operate in overlapping areas, there is competition between them. The simplest competition scheme was to consider that the extraction of n competing vessels is equal to the extraction of a single vessel divided by n. Here we considered a more complex and nonlinear approach for competition, meaning that if, for example, two vessels compete, their captures are not necessarily reduced to half, but each vessel is affected due to the interference between them. The interference is what affects the extraction of competing vessels and its extent can be calibrated. In the next section, we incorporate this parameter into the constitutive equations of the model. But here, and for illustrative purposes, we mention two extreme scenarios:

• Zero interference: The extraction rates of the vessels are independent of the number of them in the fishing site. This could model the scenario of an abundant resource;

• Maximum interference: All vessels extract the same as if they were only one vessel extracting the resource. In this scenario, the resource is scarce.

Realistic interference values are those that represent a system where vessels are not isolated from each other, but not grouped near the maximum interference situation. We achieved this by introducing an interference non-null but small parameter.

On the other hand, we modeled the influence that unregistered vessels have on the network of registered ones. Because of the uncertainty associated with the spatial positioning of unregistered fishing vessels, they were represented as a mean field of fishers that covers the entire area. We point out that illegal fishers cannot be modeled as a third strategy since, by construction, there are no “legal to illegal” or “illegal to legal” transitions in the situation to be modeled: RPA vessels are always considered legal, while unregistered or illegal vessels are only included as a uniform mean field. In this system, fishers from registered vessels have two optional strategies in presence of unregistered vessels, leading them to behave as:

• Standard fishers: they decide to ignore the vessels without RPA (the illegal fishers) and dedicate themselves to exploiting the resource. This would correspond to the cooperative strategy/behavior.

• Super fishers: they take advantage of the illegal vessels to trade with them and whitewash illegal captures with their RPA. This would correspond to the non-cooperative strategy/behavior.

It should be noted that whichever the strategy, the presence of unregistered vessels will affect the extraction rate of both players.

In order to apply Game Theory, we defined three main components: (a) the players, (b) the optional strategies of the players, (c) the utility associated with each strategy when the players compete with each other. In our model, the players are all registered vessels represented by the nodes of the network, which in turn can behave by adopting the strategy of a standard fisher or a super fisher. The players are assumed to be the leaders of the vessels who decide the strategies that will be adopted at the fishing sites. Finally, the gains associated with those strategies are represented by the extraction rates of the resource. This is, utilities in our model are not monetary figures, following previous studies (Gezelius, 2002; Gezelius, 2003; Gezelius, 2004; Hatcher et al., 2000; Kuperan & Sutinen, 1998; Sutinen & Kuperan, 1999). In this work we will use the words utility and payoff as synonyms, depending on the context in which they are used.

Figure 1 offers a schematic overview of the fishers extracting the resource with different strategies within fishing sites. Registered vessels are linked through local and variable competition intensities, illustrating indirect interference effects of super fishers on standard fishers when both are surrounded by unregistered vessels within shared fishing sites.

The agreement between a super fisher and an unregistered vessel is schematically drawn in Fig. 2, where we outline the effect of illegal fishing on the standard fisher (panel A) and the whitewashes carried out by the super fisher (panel B).

In the presence of unregistered fishers, if a registered fisher decides to behave as a super fisher, the profits of the other registered fishers are affected (reflected in a decrease in the rate of crab extraction). In other words, standard fishers with RPA lose the opportunity to extract more crabs because of the presence of the unregistered fisher. Besides, the unregistered fisher whitewashes their capture through the (registered) super fisher. The super fisher is assumed to engage in whitewashing in exchange for something (or nothing), but mainly because the unregistered fisher has no legal access to the resource and still needs to work as a fisher.

Fishing strategies and utilities

The utility of each strategy depends on the number of cooperators (the standard fishers) and defectors (the super fishers) within the pool of competing vessels. Strategy selection depends on the number of opponents in the game, the distribution of strategies among them, the mean field of unregistered fishers, the population of the king crab, and, finally, the share and interference parameters. The following equations define the payoff matrix of each strategy in this multiplayer game: (1) ckc,kd=a/n1+kc+1+ψkdγxt−1dkc,kd=a/n1+ψ1+kc+1+ψkdγxt−1

In Eq. (1), letters c and d represent the strategies corresponding to cooperating or defecting associated with a standard fisher and super fisher, respectively. Moreover, kc, kd are the number of standard and super fishers among the competitors, respectively, and xt−1 is the total population of the king crab at the end of the previous closure period. Here n is the total number of registered vessels, which in this work was set at n = 1, 000 and represents the maximum number of vessels that could hold an RPA for capturing king crab at a given moment.

Figure 1 Schematic overview of the fishing activity.

Green circles represent standard fishers whereas the red stars represent the super fishers. Green dashed lines enclose fishing areas. Links between vessels mean that they are in competition for the resource, and the blue background represents the uniform (mean) field of unregistered fishers.

Figure 2 Drawing of the agreement between the super fishers and the unregistered fishers in fishing grounds.

Boats enclosed by dotted circles represent the legal fishers and the edge represents competition between them. The crabs drawn represent the actual gain of each fisher. (A) Extraction of the registered fishers if the unregistered fisher does not act. (B) Net removal in the presence of active unregistered fishers, where the super fisher and the unregistered fisher take advantage of the standard fisher. The drawing used to depict the exchange between the super fisher and the unregistered fisher is just a simple way to visualize it, since the profit of a super fisher is not necessarily related to money.

Moreover, a is a share parameter that represents the net extraction rate of a network composed of non-competing standard fishers, that is, all that could be caught by a network of standard fishers if they did not compete with each other (this ideal case would be true if the king crab is abundant). It is worth noting that the concept of share used here does not refer to an individual transferable quota or a total allowable catch, as the fishery is not managed through quotas. However, the inclusion of this parameter in the model opens the possibility of having some control parameter of the real scenario.

Additionally, ψ represents a mean-field accounting for unregistered vessels that permeates the system. The value ψ = 0.3 was used throughout the paper unless otherwise indicated. This particular choice for the parameter mimics the real scenario since, as we said, the community of fishers estimates that around 300 fishers catch king crab without RPA. Of course, there is a lack of certainty about how many of them really are, so we chose to consider a scenario with n = 1, 000 registered fishers plus 30% unregistered fishers.

Finally, γ is defined as the interference parameter. Its role is to account for the effect of competition due to the scarcity of resources. If there is no interference, (γ = 0) the overall extraction of the n fishers involved in a competitive activity will be equal to the sum of what they extract if they were not subjected to competition. If the maximum interference is attained (γ = 1), the former net extraction will be divided by n and distributed among the competitors according to the chosen strategy and the number of unregistered fishers. Values of γ within the range (0, 1) account for intermediate scenarios. In all the simulations of this paper, we set γ = 0.25, unless otherwise indicated. As we mentioned earlier, the interference is accounting for the following effect: if the resource is abundant, competition has no effect and each vessel can extract until its payoff is complete. However, when the resource becomes scarce, competition can prevent vessels from completing this share. Even so, n fishing vessels extract more than a single one, but not n-times more. In the extreme case of high interference, these n vessels are only capable of extracting what a single vessel would have extracted.

The group of fishers is represented by an Erdös-Renyi graph (Erdos & Renyi, 1959) immersed in a mean field of unregistered fishers. The choice of this type of graph obeys the fact that a scale-free graph (ideal for the study of social networks) is indistinguishable from this simpler one due to our choice of a network with a small number of nodes. Also, the number of links was chosen in a way that every node has, on average, two other competitors; this sparsely populated network is due to the vast geographical sea space destined for the extraction of the crab. The network was generated with 1,000 links to meet these requirements.

To unravel the role of the competition parameter γ in the model, it is useful to study its effect on the profit obtained by the registered fishers. Figure 3 shows the behavior of the mean total profit of the fishers as a function of the competition parameter γ for various values of the intensities of the mean-field of unregistered fishers, ψ. Here, red lines represent the extreme case of a system composed entirely of super fishers, whereas the green curve corresponds to a system composed only by standard fishers (i.e., not lending themselves to the whitewashing activity). Each point of the curves corresponds to the mean total profit for a given value of the parameters ψ and γ. The average is performed in the steady state, on ensembles of random graphs of n = 1, 000 nodes and a fixed number of links (see the dispersion bars related to the variability of different realizations of graphs). It should be noted that the decision-making change mechanism has not yet been incorporated into these results, so the initial condition of the system in the super fisher (red lines) or standard (green line) states remains constant throughout the simulation. This result highlights the role of the γ parameter as responsible for regulating the competition/interference between fishers. High γ values imply high interference between them, and the effect of having neighbors is reflected in the decrease of the net profit of the system. Moreover, the mean total profit for these fisher’s configurations shows a smooth behavior with the competition parameter, as expected from Eq. (1). Also note that this quantity is dimensionless, since it is a combination of dimensionless quantities, and is otherwise not bounded above.

Figure 3 Mean total profit of the fishers as a function of the competition parameter.

The curves correspond to three different values of the mean field of unregistered fishers, ψ, as indicated in the figure. Each point is the average of the final state of 200 runs, and dispersion bars are included. Lines are a guide to the eye.

Choosing a strategy

In our model, registered fishers can decide to behave as a standard fishers or super fishers. Such a decision is based on the variation in the gain (or loss) in extraction rates relative to the past strategy chosen, weighed by two social parameters: the aversion to super fishing (ATSF), and the social pressure to stop whitewashing. The first parameter is related to the inner characteristics of the person who runs the vessel. The second one can have different interpretations, as it may represent the social pressure from peers, but also enforcement/surveillance pressure associated with a random control of patrol vessels that may be found on the fishing grounds.

We set the possibility of changing strategy to only one per fishing season, as it is estimated that in the real scenario the person who runs the vessel goes fishing with the decision already made. We also consider the possibility that in the next season fishers change their mind. If the relative utility is positive (i.e., standard fishers asking themselves to get a higher profit) the fisher will only change to the super fisher strategy if the threshold of the aversion parameter is reached. That is, only high aversion prevents a registered fisher from switching to super fisher in the presence of a positive utility. On the other hand, if the relative gain is negative (i.e., super fishers asking themselves to take a loss in utility by becoming a standard fisher) they will only change their strategy if the loss is less than the threshold value of “social pressure”. In other words, a super fisher (who has a positive profit) will bear the loss of acting as a standard one only if social pressure or surveillance is high enough. Equivalently, a standard fisher will become a super fisher only if the possible gain is greater than the aversion to super fishing.

In Fig. 4 we depict the possible changes in strategies for the fishers. As can be observed, the relative gain has extreme values, where the lower limit reflects the fact that a vessel cannot lose more than it is currently extracting (-100% decrease in the relative extraction rate, which in the figure corresponds to having ΔS/S =  − 1). On the other hand, the upper bound is only limited by the maximum capacity that a vessel can load on board at each fishing trip. Zero values of ΔS/S imply that the super fisher always stays as a super fisher, while ΔS/S =  − 1 means that a super fisher will always switch to the standard fisher. We estimate that in real cases this parameter takes a value between these two extremes: it is expected that there are fishers who, while being able to take advantage of the presence of illegal fishers to improve their profits, do not modify their behavior. On the other hand, there could be fishers who were super fishers at some point in the past but have ceased to be so. This type of behavior, although it may be common, is unpredictable and difficult to document.

Figure 4 Transition possibilities of the strategies used by the fishers.

The decision to change (or maintain) the fisher’s strategy is represented by regions I, II, III, and IV as a function of their relative profit gain as follows. I: The super fisher stays as a super fisher. II: The super fisher changes to standard fisher due to social pressure. III: The standard fisher remains as standard fisher and IV: The standard fisher changes to super fisher when the relative profit gain exceeds the threshold of aversion to super fishing. This scheme should be interpreted as follows. If the calculated relative gain ΔS/S of a standard fisher is positive and greater than the ATSF, then the fisher agrees to change strategy (going from standard to super fisher). On the other hand, if the calculated relative profit of a super fisher is negative—but its absolute value |ΔS/S| is less than the social pressure—then he/she accepts to change strategy (going from super fisher to standard fisher). In any other case, the fisher retains its original strategy.

In all the figures presented in this paper, the aversion parameter and the social pressure parameter are randomly chosen at the beginning of each simulation. They remain fixed throughout the process and are the same for all agents.

Fast dynamics of the fishers

In this system, the extraction rate e is calculated as the total payoff of all the vessels obtained through the payoff matrix of each individual vessel (Eq. (1)). The equation for the total extraction rates of the fisher’s system is as follows: (2) et= ∑i=1nsit

where e(t) is the net extraction rate of the system of n registered fishers at time t, and si(t) represents the payoff obtained by the i − th vessel as a result of its chosen strategy at the same time. As mentioned, according to Eq. (1), all the individual payoff of the vessels depends on (apart from other parameters) the slower time-scale crab population evaluated in the previous season time closure. This is important because it is the way of coupling the fast dynamics of fishers and the slow dynamics for the reproduction of the king crab. It is worth mentioning that our model is not the first to take into account the behavior of fishers engaging in illegal activities (see for example Charles, Mazany & Cross, 1999). There are also other works where the self-interest is taken into account as a reason for profit to engage in illegal fishing (Battista et al., 2018). Besides, in our model, the net extraction rate is hypothesized to be the entire annual capture legally landed. This is not entirely accurate because of the other forms of transgressions that are known to occur, especially gear and seasonal ban violations (Nahuelhual et al., 2018). Yet, for simplicity reasons, these transgressions are not considered here.

Slow dynamics of the king crab population

Together with the dynamics of the fisher’s activity, we also include that associated with the king crab population. Each of these dynamics is associated with its own characteristic time such that when they are compared with each other, the dynamics of the king crab population is much slower. In our model, we suppose that king crab population is uniformly distributed. This assumption is mainly due to the scarcity of data regarding the spatial dynamics of the species in a region as extensive as Magallanes. According to official reports conducted by Chilean technical agencies, there are limited community and environmental studies associated with king crab in the region. Existing king crab community studies consist of an image of the communities and do not account for the ecological interactions or the temporal dynamics associated with them. The same happens in relation to the possible effects of disturbances (e.g., illegal fishing) reported as relevant (de Arauco, 2019).

Regarding the population dynamics of the crab, while no specific studies of population regulation were found for this species, positive density dependence has been observed for different crustacean species in marine ecosystems because of natural and human influences (Gascoigne & Lipcius, 2004; Gascoigne et al., 2009), including the related red king crab in Alaska (Zheng, Murphy & Kruse, 1995). Thus, we follow the ideas expressed in 1932 by Allee & Bowen (1932), who suggested the possibility that individuals in a population can benefit from the presence of conspecifics, implying that in some cases instead of intra-competence there could be positive feedback, which could be crucial for the survival of a species. The phenomenon called the “Allee effect”, does not have a clear definition. One of the most accepted interpretations and the one we follow here points to the difficulty experienced by the individuals under study to mate when the population density falls below a certain level. The Allee effect can also be present in situations in which the benefits of con-specific presence may include dilution or saturation of predators, surveillance and defense, and cooperative predation, among others. The Allee effect is then a positive association between individual aptitude and population size. Such a positive association may result in a critical size below which the population cannot persist (Stephens, Sutherland & Freckleton, 1999).

The equation for the dynamics of the king crab has two terms, the first that corresponds to the Allee effect and the second that accounts for the extraction due to the fisher’s activity: (3) dxdt=x1−xx−λ−exx0Hsin2πt.

This dynamic was proposed as a realistic model for the species due to the fact that it needs a minimum threshold population (which in the real scenario is unknown due to lack of research in that field) to reproduce until reaching the carrying or saturation capacity of the habitat. If the population falls below this threshold value, the remaining population is not able to sustain the fishery. In our model, and in the absence of extraction, this critical king crab population threshold value is represented by λ = 0.1, i.e., when the population falls below 10% of the maximum king crab population. This value was used throughout the paper (unless otherwise indicated) and chosen as a reference because values close to it were reported when the Alaska king crab fishery collapsed in the early 1980s (Dew & McConnaughey, 2005b).

Besides, in Eq. (3) the parameter of interest is the ratio of the extraction rate to the king crab reproduction rate. This ratio will be called the “relative extraction rate” or simply “extraction rate” (e). Besides, H is the step function. The time unit is such that, in one year, there is a fishing/closure period of 6 months each. This selection, without loss of generality, is an approximation used in the model for mathematical and computational convenience, whereas the actual scenario is usually 7 and 5 months, respectively. In the following section, we also present scenarios with extended ban periods.

Model structure

Our model can be thought of as composed of three interacting layers. Each of them involves one or more components of the system that have their own dynamics and scale of description.

One layer represents the king crab habitat, which we assumed as being uniformly distributed and that has a dynamic behavior given by the Eq. (3).

A second layer represents the mean field of unregistered fishers which is also assumed to be uniform since it is not possible to know which vessels are whitewashing at a given time and where they are located. The effect of illegal fishing on the system is incorporated into the Eq. (1) through the parameter Ψ.

Finally, the third layer corresponds to the network of registered fishers distributed throughout the fishing sites. Fishers are connected by weighted links forming an Erdös-Renyi random graph. Registered fishers define their strategy following Eq. (1) and considering the social parameters depicted in Fig. 4. As a result of their decision, an extraction value given by Eq. (3) is obtained. This value is used to feed Eq. (2) which in turn allows calculating the population of the king crab that will be available in the next fishing period.

Effect of extraction rates on king crab population

Here we analyze the theoretical framework for the evolution of the crab population in the hypothetical scenario of a fixed extraction rate. In Fig. 5A, we present the phase portrait of the slow dynamics linked to king crab population, which we denoted x (see Eq. (3)). We also include, in panel B, the bifurcation diagram of the system, since it is useful to identify the regions of values for the parameters needed for the maintenance of the fishery i.e., asymptotic non-zero values of the king crab population. There is a threshold value for the extraction rate e∗ that separates the regions of conditional sustainable exploitation, i.e., avoid the exhaustion of the resource, (regions from I to III) from the unsustainable fishing activity (region IV). This means that if a critical extraction rate is exceeded, the system will always collapse, while in other cases the sustainable extraction of the fishing activity depends on both the extraction rate and the value of the king crab population.

Figure 5 Phase-portrait and bifurcation diagram for fixed extraction rate.

(A) Phase-portrait of the normalized king crab population for increasing values of the extraction rate, e = 0, 0.14, 0.2025, and 0.25 (from the black curve to the red one). (B) Bifurcation diagram of the system with explicit regions of sustainable fishing (I, II), and unsustainable fishing (III, IV). For λ = 0.1, the critical extraction rate is e∗ = 0.2025.

It is important to highlight that the king crab dynamics explicitly depends on the Allee parameter λ (which we set to λ = 0.1), and on the relative extraction rate e, according to Eq. (3). Then, the critical value where the bifurcation occurs is in a vertical line passing through a threshold value e∗.

From Eq. (3) the critical values of the crab population x± that separate region II from regions I and III (which are the stable and unstable critical points, respectively) can be calculated as a function of the extraction rate and the Allee parameter. Setting to zero the right-hand side when H = 1 and discarding the root x = 0 we obtain: (4) x±=1+λ2±121+λ2−4λ+e

It is important to note that a direct relationship between e∗ and λ is obtained from Eq. (4), by equating the critical values x+ = x− (when the saddle–node bifurcation occurs). This gives us the expression e∗=141+λ2−λ.

This equation is relevant because it would allow empirically inferring the value of the crab Allee parameter, estimating the region of sustainability in which the fishery is located. One could even go further and try to fit the region around the vertex of the parabola to get a better approximation of λ. In other words, it follows from our model that if this Alle parameter exists (which we believe it does, given the biological necessity for a critical minimum population to exist) then its value is related to the critical extraction rate.

As we said, the slow and fast dynamics are coupled in a way that at the end of the fishing season, the new value of the king crab population is calculated from its own dynamics, according to its reproduction rate without extraction. At the beginning of the next fishing season, the new value for the king crab population is used to calculate the new profits in the payoff matrix (Eq. (1)).

At this point, it becomes necessary to briefly discuss what we mean by sustainable or unsustainable fishing in the context of this model. We will say that the fishery is sustainable if the rate of extraction relative to the rate of reproduction of the crab (which we define in Eq. (3) and named e) is such that in a reasonable time the population of king crab, when it is not being captured, recovers to a certain number that warrants its survival as a species. As we will see in the next section, in our model this means that the stationary king crab population oscillates around fixed values but never reaches extinction.

Results

We begin this section by presenting the results of the dynamics observed in one of the layers of the model, specifically, the one corresponding to the king crab, assuming a constant and fixed extraction rate. The analysis of these results will allow us to understand how this layer is connected to that of the network of fishers, a problem that we address in a later section.

Net extraction rate and king crab dynamics

In this section, we analyze the coupling between the king crab dynamics and the net extraction rate of fishers, for fishing season periods of half a year and a mean field of unregistered fishers of ψ = 0.3. Understanding this value for the mean field is useful to analyze what the extreme value of ψ = 1 means. The value of ψ = 1 implies that, on average, there are as many unregistered vessels as registered ones, in the hypothetical scenario where registered fishers are in zero interference.

Figure 6A shows the time evolution of the king crab population during 50 years for two single runs with different values of the share parameter a. For a high value a = 0.83 we obtain an extraction rate e = 0.44 and the king crab population collapses, as expected. This can be observed in Fig. 6B, where the evolution is now plotted in the bifurcation diagram. However, a different situation is obtained for a medium share value a = 0.64, which results in an extraction rate e = 0.38. In this case, the king crab population stabilizes at 68% of the maximum population, which can be interpreted as an example of sustainable fishing. However, according to the observed in Fig. 5B this value of extraction rate should lead to unsustainable fishing since it belongs to region IV of the bifurcation diagram. This apparent contradiction occurs because fishing takes place only during the fishing season and not across the entire year, allowing the king crab population to recover from exploitation. This result brings motivation for the model to introduce a dynamic where the length of the fishing season is an important parameter to consider.

Figure 6 Evolution of the king crab populations in sustainable/unsustainable regimes.

(A) King crab population as a function of time for a medium value of the share parameter (a = 0.64, blue line), and a large value (a = 0.83, red line). (B) King crab population as a function of time in the bifurcation diagram view for the same runs of panel A. Extraction rates at year 50 are e = 0.38 (blue dots) and e = 0.44 (red dots). All the realizations were prepared for the worst case scenario with initially all the fishers being super fishers. The numbers in parentheses represent the values of e and the final king crab population, respectively.

Net extraction rate and social parameters

The results of the previous section were obtained for a single realization of the model, which implies having used single values of the social parameters. We present here the results of several simulations with different values of social pressure and aversion to super fishing.

In Fig. 7A, we show the evolution of realizations with random values of the social parameters and several values of the share parameter a, as indicated in the figure. All other parameters are the same as in Fig. 6, including the initial condition, which corresponds to preparing the system with all super fishers. In this scenario, the evolution for a 50 years span for the joined (slow and fast) dynamics corresponds to a non-zero king crab population above 50% for some of the realizations. This means that, even in this unfavorable situation of intense extraction, it is possible to have sustainable fishing of the king crab. The possibility of having a sustainable scenario in this model is achieved with an appropriate combination of parameters. In this case, high values of aversion to super fishing and/or intense social pressure, properly combined with the share parameter, can help avoid the collapse of the fishery. This is also observed in Fig. 7B, where we plot the final values of the king crab population as a function of the share parameter a. The king crab population in the stationary state takes values greater than 50% for values of the share parameter significantly high. Moreover, three distinct behaviors for the king crab population as a function of the share parameter a are observed in Fig. 7B. In all three cases, however, there is a critical value of a above which the collapse is unavoidable.

Figure 7 Temporal evolution of king crab populations and final states after 50 years.

(A) Evolution of the king crab population as a function in time for 30 runs with random values of the share parameter a and of the social parameters. The color scale indicates the value of a in each simulation, where the maximum (minimum) value of a corresponds to the lower (upper) curve. The arrow indicates the increasing direction of a. (B) Final values of the king crab population versus the share parameter for 500 runs. In all the simulations, the initial condition involves only super fishers.

The branches observed Fig. 7B correspond to three global dominant strategies adopted by the fishers when the system converges. This can be also observed in Fig. 8A, where we plot the number of standard fishers as a function of time for several sets of parameters chosen at random. It is worth noting that the evolution of the system has only three possible behaviors, despite the fact that the parameters are different in each of the 500 simulations carried out. Interestingly, although in all the runs the fishers are initially super fishers, there is an emergence of standard fishers in the system for some sets of parameters (as is the case of orange and green curves). Additionally, Fig. 8B was constructed with the final states of the runs of Fig. 8A and shows that the extraction rate e and the share parameter a are related, as we should expect, but certainly are not the same. Also, the result shown in this figure is connected to the behavior of the king crab population of Fig. 7B by the share parameter through the convergence of the global strategies adopted by the fishers in Fig. 8A.

Figure 8 Combined effect of the parameters on the final state of the system.

(A) Evolution of the total number of standard fishers in the system for random values of the share parameter a and of the social parameters. (B) Total extraction rate of the registered fishers at year 50 as a function of the share parameter values. In both figures, 500 simulations were run with the same initial conditions as in the previous figure.

Extraction rate, social parameters, and fishers strategies

In our model, the dependence of the extraction rate on the share parameter is explicitly given through the payoff matrix (Eq. (1)). The results of the previous section showed that there is a dependence of the asymptotic values for the extraction rate e with the share parameter a. It is then important to know how that dependence manifests itself. The three different behaviors observed in (Figs. 7B, 8A and 8B) can be understood if we plot the number of standard fishers in the steady state as a function of the two social parameters, as we did in Fig. 9. The figure shows three different regions in the plane given by the aversion to super fishing and the social pressure. When the absolute value of the social pressure is low (red region), the stationary state corresponds to all fishers acting as super fishers, that is, the final state coincides with the initial one. When the absolute value of social pressure is higher than a threshold, standard fishers emerge. In this case, for low values of aversion to super fishing a region with half the population following each strategy is observed (in orange), while the widest green region with higher values of aversion to super fishing corresponds to all fishers being standard.

Figure 9 Number of standard fishers in the stationary state as a function of the social pressure and the aversion to super fishing (ATSF).

Red downward triangles (rightmost region of the figure) correspond to a final state composed only by super fishers. Orange rightward triangles (bottom region) indicate final states where half of the fishers are super fishers and half are standard ones. Green upward triangles are final states composed exclusively of standard fishers. As in the previous figures, the initial condition involves only super fishers.

The lines separating regions are easily obtained from Eq. (1). The change of profit ΔS to go from standard to super fisher is ΔS = d(kc, kd) − c(kc, kd). Dividing this expression by the initial profit, which in this case is the standard one, S = c(kc, kd), gives ΔS/S = ψ. Similarly, to go from super fisher to standard, ΔS = c(kc, kd) − d(kc, kd) and the initial (super fisher) profit is S = d(kc, kd). Then, we obtain ΔS/S =  − ψ/(1 + ψ).

The horizontal line in Fig. 9 indicates the threshold to go from the standard to the super fisher strategy. If ΔS/S = ψ is greater than the aversion to super fishing, then the temptation is greater than the threshold and the change will be accepted. On the other hand, the condition to switch from super fisher to standard fisher is ΔS/S =  − ψ/(1 + ψ). If this loss is considered small (that is, less than the social pressure), the change of strategy is accepted. This is the vertical line of Fig. 9. The decision to accept or reject the strategy change based on the relative gains and social parameters is made as represented in Fig. 4.

It is worth noting that the stationary state of the system is independent of the aversion to super fishing when the social pressure in absolute value is less than the expected relative loss |ΔS/S| = ψ/(1 + ψ) (red region of Fig. 9). This is foreseeable because social pressure just plays a role when the only possibility is to change from standard fisher to super fisher. In this region, the steady state remains the same as the initial state (all super fishers).

When the absolute value of the social pressure is higher than the threshold, the standard fishing strategy emerges (green and orange regions) as the super fishers are now forced to change their strategy. Both regions have a non-zero number of standard fishers and are separated by the other threshold ψ.

The green upper left region of Fig. 9 can be understood as follows. Since the social pressure (in absolute value) is higher than |ΔS/S| = ψ/(1 + ψ), then (according to Fig. 4) the super fisher is forced to take a loss and change to the standard fishing strategy. Afterward, if at a later time the same fisher (now with the standard strategy previously selected) is chosen, they can not go back to the super fisher strategy, as in this region the aversion to the super fishing threshold is higher than the expected relative gain ΔS/S = ψ. Consequently, the final state is a full consensus of standard fishers. This is also the reason why the green line of Fig. 8A reaches the maximum value in the asymptotic regime.

Finally, the orange region of Fig. 9 can be analyzed in a similar way to the previous case. When standard fishers are selected, they must change to super fisher strategy because in this region the relative profit gain ΔS/S = ψ is greater than the aversion to the super fishing threshold. On the other hand, since the absolute value of the social pressure is higher than |ΔS/S| = ψ/(1 + ψ), a super fisher is forced to change to the standard fishing strategy. In conclusion, both kinds of fishers will change to a different strategy other than the present one and, in the mean, the number of standard fishers converges to half of the total population (see also the orange line of Fig. 8A).

On the role of the fishing season duration

In this section, we study the dynamics of the king crab-fishers system allowing a varying closure lapse. We are aware that from a biological point of view, increasing the fishing period is not sustainable. Also, we know that restricting the fishing period has economic implications for the fishers, being socially and politically unfeasible. But given the advantage of the computational model, we can explore the consequences on the king crab population of changing the fishing season duration.

The equation for the dynamics of the king crab (Eq. (3)) was modified to include a new parameter that accounts for the duration of the extraction season inside one year. The new equation for the king crab population with variable season times is: (5) dxdt=x1−xx−λ−exx0Hsinπt−Hsinπt−σ,

where σ represents the fraction of the year corresponding to the fishing period with respect to the duration of one year. This new equation recovers the previous situation if we set σ = 0.5.

In Fig. 10A, we depict the implementation of the parameter σ as the regulator of the opening and closing of the fishing seasons. In Fig. 10B we show how the evolution of the king crab population would behave from the Eq. (5) if the extraction rate e remains fixed and set to 1, but the duration of the closure 1 − σ changes. These curves are purely theoretical, as the connection to the fisher’s network has not yet been taken into account. We see that, for zero extraction (σ = 0), the king crab population remains constant at its maximum value. This is, by definition, what happens when the king crab population saturates at the carrying capacity of the system, which depends on the biology of the species and the number of resources available for its reproduction, and there exists no extraction whatsoever. It is important to mention that, as we comment before, the actual value of this maximum population is unknown. However, for our purposes, we can set it to be equal to 1 without loss of generality. The increase in the duration of the fishing season produces a decrease of the king crab population. The higher the value of σ, the steeper the decrease in the king crab population over time. In the extreme case where the resource is extracted throughout the whole year, (σ = 1), the king crab population collapses in a very short time (∼ 4 years). This steep decline in the crab population is due to the fishery remaining open all year long and that the extraction rate e = 1 is far away in the right side of Fig. 5B (region IV of the bifurcation diagram).

Figure 10 Fishing season duration σ and evolution of the king crab population.

(A) Representation of the opening and closing of the fishing seasons. (B) Time-evolution of the king crab population for several values of the fishing season duration σ, as indicated in the figure. The curves are obtained from Eq. (5).

It is important to highlight that, in the real system, all these periods and behaviors are closely related to the biology of the king crab, the resource available for its reproduction, the extraction rate relative to its reproduction rate, and the minimum threshold population λ below which the system would collapse due to the low density of specimens (even if the resource is no longer extracted). Therefore, in order to have a realistic time scale, it is necessary to calibrate all these parameters based on what is known about the species. However, the qualitative dynamics of our model and the conclusions regarding the expected behaviors (collapse of the king crab for high harvests, very long fishing seasons or very high field values of fishers without registration, and recovery of king crab for stricter shares or for longer closed seasons, among other things), still holds regardless of the lack of detailed, empirical and realistic parameter values.

The incorporation of the parameter σ in the complete model gives rise to the results shown in Fig. 11, where we plot the state of the king crab population after 50 years, for different values of share and duration of the fishing seasons. We can see, unsurprisingly, that for higher share values and for very long extraction seasons, the population collapses to zero. We also observe that even though the duration of the fishing season was introduced in the crab population dynamics (Eq. (5)) and the share parameter was introduced in the fisher payoff matrix (Eq. (1)), both parameters have approximately the same impact on the stationary population values of king crab. This is reflected in the symmetry of the results, which is highlighted by the dotted line we draw in the figure. This fact can be interpreted as advantageous since modifying the duration of the fishing season or modifying the shares may be either easier or more difficult depending on management agreements. Besides, the king crab population changes abruptly when approaching a threshold region that depends on both parameters, σ, and a. However, the intrinsic fluctuations of the real system could modify this scenario and lead to unwanted population collapses, so a management decision based on these results should consider staying away from that frontier.

Figure 11 King crab population at year 50 as a function of the share parameter and duration of fishing seasons.

Each point is the final state of one of the 1,700 simulations shown in this figure. The color bar indicates the value of the king crab population in the steady state.

Finally, Fig. 12 shows a different approach to the previous results. Here we analyze the state of the system at a medium term of 50 years in the bifurcation diagram, to help/guide management. We can see that the introduction of the σ parameter generates the emergence of a sustainable fishery in regions of the theoretical bifurcation diagram in which unsustainability was expected (regions III and IV in Fig. 5B). This is in agreement with the results shown in Fig. 11, where this behavior is due to the duration of the fishing season. The colors represent different durations of fishing seasons, that is, different values of σ.

Figure 12 King crab population in the bifurcation diagram.

Symbols represent the population after 50 years. For each of the 1,700 runs, random values of the share parameter a and the fishing season duration σ were selected. The color bar indicates the value of σ. Lines are the bifurcation diagram of Fig. 5B.

In summary, not only the extraction rate (through the share parameter) is important to preserve the sustainability of the resource, but also selecting properly the season length. Our results indicate that the sustainable fishery can be recovered for high extraction rates, a problem that by hypothesis is difficult to control, if the duration of the fishing season is chosen appropriately. However, this can only be achieved if all fishers respect the fishing season. If, on the other hand, they decide to behave selfishly and fish out of season, the tragedy of the commons occurs, collapsing the fishery (sigma points close to one and stationary crab population close to zero). This can be interpreted as effectively using a longer fishing season than previously agreed upon.

Discussion

In this work, three models are coupled to analyze the problem of illegal fishing and its consequences on the sustainability of the resource. Various studies have been carried out on partial aspects of the system from a global scale (Agnew et al., 2009; Bianchi et al., 2014; FAO, 2016), where they estimate the extent of illegal fishing and its threats, down to the local scale (Cavole, Arantes & Castello, 2015; Luomba, Chuenpagdee & Song, 2016). Ethical aspects of illegal fishing are discussed in Satizábal et al. (2021), and it is even justified (Cepić & Nunan, 2017) to cite a few examples.

Our work provides a modeling basis to mathematically characterize a complex system composed of several interacting layers that have their own dynamics. These layers reflect diverse subsystems and include characteristics as different as the biology of the crab, the motivations for deciding to be a super fisher Battista et al., 2018, or its consequences on sustainability Hauck, 2008; Nahuelhual et al., 2020.

Taking into account the behavior of the parameters that describe these phenomena, and combining the prisoner’s dilemma of game theory (Berkes, 1987; Borda, 2011; Cole, Izmalkov & Sjöberg, 2014) with the crab dynamics equation that includes the Allee effect (Allee & Bowen, 1932), our model provides elements for the management of the fishery and of the commons in general, taking as its starting point the pessimistic hypothesis of Hardin (1968) about human behavior when they have no control in the exploitation of shared resources.

From a game theory point of view, this research fills a knowledge gap as we use this framework to model fishers’ decision-making to take advantage (or not) of the presence of illegal fishers. This decision is also influenced by social parameters that modify the behavior of the fishers. On the other hand, the spatial distribution of legal fishers is taken into account in the structure of the network. This is relevant from the point of view of the game since a different spatial distribution would change the structure of the network (changing the competing neighbors kc, kd in the Eq. (1)) and therefore would modify the dynamics of the game and its outcome.

Main findings and their meaning

One of the main findings of our work is that by using a model for the king crab dynamics that includes the Allee effect (Allee & Bowen, 1932) and an extraction term due to fishers’ activity, we can determine zones of sustainability or unsustainability in resource extraction. The Allee parameter λ is a critical threshold of a hypothetical population that, even in the absence of extraction, cannot recover and the resource collapses.

The game theory model combined with the model for the king crab dynamics allowed us to explore answers to the main questions raised:

• Our model shows that individual benefit will be maximized with individualistic behavior, where all registered fishers behave like super fishers. This situation persists for a wide range of values of the social parameters (red region in Fig. 9).

• However, which behavior is more likely to prevent the decrease of the king crab population under a critical threshold as postulated in the tragedy of the commons? Our model shows that individualistic behavior is unsustainable and leads to the collapse of the fishery (Figs. 7 and 11), as predicted by the tragedy of the commons (Hardin, 1968). It is important to highlight that the cooperative strategy, that is, behaving like a standard fisher ignoring the advantage offered by illegal fishers, is a necessary condition for sustainability but not sufficient because sustainability also requires a lower net extraction or shorter duration of fishing seasons.

The model allowed us to analyze under what conditions the trade-off between individual and collective benefits works, finding that the condition of unsustainability ends, sooner or later, also deteriorating individual benefits. It is observed in the payoff matrix that the individualistic behavior (choosing the super fishing strategy) is always better to maximize the extraction rate than the cooperative behavior (i.e., behaving like a standard fisher), but the high values of the total extraction rate of individualistic behavior, being far from the bifurcation point (Fig. 6), is what leads to the collapse of the fishery, a result that is neither beneficial for the system nor for the individuals (Hardin, 1968).

It is important to clarify that our model assumes the presence of illegal fishing in the system but does not seek to explore ways to solve it, but rather to understand the conditions under which the interaction between legal and illegal fishers may or may not negatively affect the king crab population. Based on our knowledge of the system in the Magallanes region, we know that the whitewashing agreed between registered fishers in presence of unregistered ones, is a practice as old as the existence of the access prohibition given by the RPA, which has been maintained over time and it has become naturalized not only among fishers but within the entire value chain (Nahuelhual et al., 2020).

For this reason, the practice of working together with unregistered fishers regardless of whether this breaks the law does not necessarily imply a long-term social optimum. This is because the option that maximizes short-term utility (the strategy followed by the super fishers) admits the presence of unregistered fishers but also the choice of whitewashing, which is a sanctioned activity. This is one of the main reasons why this game is novel, as it differs from most game theory applications to illegal fishing in SSF, where interactions between illegal and legal fishers are not assumed to exist.

In cases of overfishing, understood as unsustainable fishing, which constitutes most of the applications of game theory and of the Prisoner’s Dilemma in particular (Byers & Noonburg, 2007; Bellanger et al., 2019; Davis & Harasti, 2020; Grønbæk et al., 2020a; Grønbæk et al., 2020b; Grønbæk et al., 2020c; Grønbæk et al., 2020d; Kaitala, 1986), it is easy to show that the option of cooperating (not overexploiting) is superior to the option of defecting (individually extracting as much as possible), as stated by Dietz, Ostrom & Stern (2003). However, in our game, neither of the two options is really optimal from the point of view of normative regulation, yet it is a case that merits much attention in SSF. Beyond the evidence that exists or not regarding the effects of illegal fishing on the dynamics of a species (Satizábal et al., 2021; Gezelius, 2002; Gezelius, 2003; Gezelius, 2004; Hatcher et al., 2000), the authority, from a regulatory and criminal point of view, has the moral responsibility to act on normative transgressions. Whether or not the sanctioning route, the deviantization and criminalization of the fisher is a solution to the problem, goes beyond what we analyzed in this study.

The model is therefore valuable to represent SSF systems with the presence of illegal fishing, which is highly common in fisheries around the world, especially in developing countries (Luomba, Chuenpagdee & Song, 2016; Nahuelhual et al., 2018; Nahuelhual et al., 2019; MacKeracher et al., 2021; Okeke-Ogbuafor, Gray & Stead, 2020) and to reflect some aspects of the real system such as:

1. Legal fishers, in general, admit whitewashing and some even collaborate (the super fishers) either for profit or just out of solidarity with those who do not have RPA.

2. Whitewashing and super fishers are naturalized within the system without apparent conflict, given the perceived abundance of the species; our results indeed suggest that reasonable kings crab population levels can be maintained even in the presence of illegal fishers and super fishers.

3. The generality within the system is the presence of some super fishers and not many of them. Although social pressure plays a role in controlling this behavior, it should not be forgotten that the super fisher, when it is a carrier boat, is also the one who buys the crab from fishers with RPA who certainly have no interest in reporting the super fisher with whom they maintain commercial ties.

4. Despite this system dynamic, the king crab has not collapsed. However, the species has been in a regime of full exploitation since 2014 and with a temporary suspension of the RPA since 2019, as precautionary measures, given the falling trend in landings.

How would it be possible to promote cooperation to keep king crab from collapsing?

The introduction of the social pressure parameter in the model leads to the appearance of cooperative behavior, although no additional gains are obtained. The model answers the question posed above by suggesting that some form of social control needs to be applied, either top-down (from law enforcement authorities) or bottom-up (among fishers) to eradicate or diminish in the system the behavior of the super fisher.

We found that there are a range of values for this parameter, together with the others defined in this work, within which king crab would not collapse. The way to enforce the law would be more direct control of the fishing grounds, which is very challenging given the size and geography of the Magallanes region and the limited resources of the fisheries authority. Another possibility is that formal or informal social control comes from consumers, both local and international, through market agreements such as certification.

A different top-down manner to regulate the system to maintain sustainability is through the σ parameter, which represents the duration of the fishing season. A similar result is obtained by applying a share a for the extraction of the species, however, it is more difficult to control than σ. This is an important finding for management since they are two parameters with a similar impact, but in real life, they can be very different in terms of the socio-political viability of their application.

In the scenario of intermediate values for the extraction rates and shortening the extraction period (longer closures), a collapse can be prevented. Comparing the model with real life, the extension of the ban or reducing the extraction rates can be harmful measures for fishers. But, be that as it may, in the long term the collapse of the fishery would be even more damaging. Under current management, the only way to avoid the tragedy of the commons is to extend the closure period, or even close the fishery altogether. Decreasing extraction rates implies costly monitoring of fishing effort (i.e., number of traps), which is very difficult to implement, while quotas would also have to be introduced and controlled since the fishery is not currently managed with individual quotas.

Government representatives acknowledge the seasonal ban as likely the only measure that could be implemented to control overfishing arising from different transgressions, but they also recognize the negative effect it could have on fishers’ income.

Since our exercise is modeling, it is important to note that it is not possible to suggest precise numbers of the control parameters. To obtain this precision, information is needed on parameters such as Allee’s or the reproductive rate of the species, as well as information on the spatial arrangement of the population (knowing whether it is uniform or not). Having modeled the parameters that lead to sustainability in a theoretical way, our model opens the possibility of deepening research in this line, increasing precision if the information that is needed is obtained.

Critical king crab population value and the Allee parameter

It is interesting to note, for future research, that Fig. 12 reproduces very well the curve in the bifurcation diagram obtained under the hypothetical fixed extraction rate scenario. If we solve our model for different values of the Allee parameter (λ) and then fit the corresponding parabola making use of the real or estimated data for the extraction rate we could, in principle, be able to obtain a realistic estimation of the Allee parameter for our model.

With this parameter at hand, making use of the extraction rates, and remembering that the extraction rate e is the “fishing extraction rate relative to the reproduction rate of the king crab”, the Eq. (4) could be a fitting equation to estimate the actual, unknown, reproduction rate of the king crab. This estimated parameter is highly difficult to obtain due to the lack of knowledge of the species unless a biological study is carried out in the laboratory and therefore this could be an indirect way of obtaining it.

Implications for management: How does each strategy affect king crab population in the long-term?

The super fisher strategy leads the king crab stock to fall under a critical threshold, as postulated in the tragedy of the commons hypothesis, under the following conditions:

1. High values of the share parameter a.

2. High values of the duration of the fishing season σ, which presupposes that a shorter period avoids collapse regardless of the number of unregistered fishers, assumed fixed in the context of our model, and super fishers.

3. High values of the mean-field of illegal fishers (ψ, taken as fixed in our work). The interaction between many unregistered fishers and super fishers lead to collapse.

4. High/low values of social pressure. The social pressure parameter is justified on the assumption that a strong negative pressure towards super fishers can exert some bottom-up control. This pressure can come from their peers or from their community. Nonetheless, our model shows that even with high social pressure there can be a high whitewashing activity between illegal fishers and super fishers.

Impact of the social parameters in the behavior of the fishers strategies

We mentioned that the sustainability of the fishing activity is (trivially) strongly dependent on the exploitation rate of the resource. However, in our model, the extraction rate is, in a certain sense, a variable that is out of the control of the fishers since it cannot be modified as a result of a management decision, but it is also true that the extraction rate explicitly depends on the share parameter through the payoff matrix (Eq. (1)).

Besides, in the context of game theory, all fishers are rational players, since they will only change strategy if their payoff is higher than before. In order to introduce the non-rational behavior, i.e., change strategy when accepting some loss in payoff, we introduced the social parameters. They are the cause of the emergence of standard fishers even in an initial setup of all of the registered boats with the super fisher strategy and having permeated with a non-zero mean field of unregistered fishers.

The aforementioned emergence of standard fishers becomes clear in Fig. 8, where we find three values of the extraction rate for each value of the share parameter. The emergent quality of three distinct behaviors for our system is linked with the social parameters. In the plane of social parameters presented in Fig. 9, the stationary states are partitioned into three different regions, and the thresholds that separate them are functions of the mean-field of unregistered fishers, ψ.

Model limitations and future research

One of the main limitations of our model is that there is no direct interaction between the illegal fishers subsystem and the crab dynamics. Therefore, it is not possible to evaluate, for example, the effect that the opening of the RPA would have, that is, “the legalization of illegal fishers”.

In this model, for simplicity, a uniform spatial distribution of the king crab was used. In future work, a random and/or non-uniform distribution of the resource could be chosen. Related to this, a crab protection area could also be introduced into the model that can serve as a source of resources to the fishing grounds, incorporating flow parameters, propagation speed, or reproduction rate within it.

Another limitation is related to the Allee parameter and the lack of knowledge about the species. In our model, it is hypothesized that there is a minimum population value needed in order for individuals to encounter and mate, but we relate this quantity to the critical value of e from which region IV begins in Fig. 5B.

Besides, our model does not include a parameter such as catchability, which would be interesting as a complement for future work. This parameter would affect the extraction rate since not everything that fishers intend to capture is currently captured. It would then be an extra interaction of the registered fishers with the crab population and its effect would be that for non-ideal catchability, there would be an e “effective”, necessarily less than (or at most equal to) our current e. With this inclusion, we could answer other types of questions different from those we pose in this work, such as: given the catchability found experimentally, what would be an estimate of the current population of crabs?

We are aware that a more realistic version of the model could include a link between prices and crab dynamics, which would affect the rate of extraction of the resource. However, we have not incorporated this mechanism into the model for the following reasons:

1. Chile is a very small player in the global market of crabs, and competition between fishers in a certain region of the country does not affect international prices that are set by companies before the fishing season begins (in other words, Chile is a price taker). Most king crab catches are exported, while the small quantities that are marketed locally come from unauthorized and artisan practices. This does not rule out the possibility of modeling a scenario where prices fluctuate, but it should not be done in response to competition among fishers.

2. The effect of competition on prices is also limited by the trade structure. For the most part, crab exchange is possible or facilitated by enabling mechanisms, which are very common in all Chilean and Latin American SSF. In this way, fishers obtain the supplies (bait, oil, food, money) necessary for the fishing operation, activating a debt that the boat owner must pay with the captured catches and discounting the money or resources advanced. Through these enabling mechanisms, the price is imposed by the processing plant or by the intermediary, whereas the boat owner does not have the ability to negotiate prices or establish agreements with their peers to increase profits. The indebtedness causes a race to get a catch at any cost and thus pay off the debt and make whatever profit possible.

For future research on this topic, more questions could be considered:

1. How will the model respond if the fishery opens the RPA?

2. How will the model respond if we introduce a non-uniform distribution of king crab in the fishing grounds?

3. How does the share parameter impact the behavior of unregistered fishers?

4. How would the total extraction rate of the resource respond to the presence of patrol boats that could influence the decision-making of fishers through some type of sanction for super fishing?

These types of questions could open the possibility of increasing the reality of our model and also introduce more control variables to the fishery.

Concluding remarks

Our work shows that the interactions between legal and illegal fishers and their effects on the fishing resource can be modeled through game theory (Prisoner’s Dilemma) coupled with a dynamic model of the crab population. The competition of the fishers was modeled using a non-directed weighted network and the coupling of this system with the problem of illegal unregistered boats for extracting the resource was included as a mean-field model for the intensity of the illegal activity. The influence of this field on the proper dynamics of competition among the registered boats was presented in the context of game theory of multiplayer games where the payoff was the local extraction rate of king crab relative to the reproduction rate of the species.

We define parameters that could help/guide management to deal with/minimize the problem of illegal extraction with the sustainability of the fishing activity in mind in order to avoid the collapse of the fishery.

Similar to overexploitation models, the sustainability of the fishery strongly depends on the rate of harvest. The extraction rate, in turn, depends on the number of illegal and super fishers.

By construction of the payoff matrix, the super fisher strategy is always better at maximizing the gains than the standard cooperative strategy, but both strategies have different gains only at non-zero values of the average field of unregistered fishers.

The extraction rate is beyond the control of fishers, therefore it is a variable that must be managed from a top-down central management decision.

If, due to logistical limitations, it is not possible to avoid illegal extraction, an effective alternative would be the establishment of extraction quotas per vessel and a monitoring system that ensures compliance. In terms of the parameters of the model, this implies finding a way to indirectly reduce the net extraction rate of registered fishers through an appropriate selection of the share parameter and, on the other hand, increasing the social pressure that makes the behavior of standard fishers emerge.

Social pressure and aversion to super fishing are two parameters that symbolize bottom-up control, which combined with top-down regulations, can have a positive effect on king crab population.

In conclusion, our model indicates that the collapse of the fishery would only be avoidable with a combination of top-down and bottom-up decisions, since neither of the two, by itself, would be sufficient.

We especially thank the editor and the two rounds of reviewers that this paper had. Thanks to the meticulous revision work our work improved a lot since the first version.

Additional Information and Declarations

Competing Interests

Author Contributions

Data Availability

The authors declare there are no competing interests.

Alan Zambrano conceived and designed the experiments, performed the experiments, analyzed the data, prepared figures and/or tables, authored or reviewed drafts of the article, and approved the final draft.

María F. Laguna conceived and designed the experiments, analyzed the data, authored or reviewed drafts of the article, and approved the final draft.

Marcelo N. Kuperman conceived and designed the experiments, analyzed the data, authored or reviewed drafts of the article, and approved the final draft.

Pedro Laterra analyzed the data, authored or reviewed drafts of the article, and approved the final draft.

Jorge A. Monjeau analyzed the data, authored or reviewed drafts of the article, and approved the final draft.

Laura Nahuelhual analyzed the data, authored or reviewed drafts of the article, and approved the final draft.

The following information was supplied regarding data availability:

The code is available at GitHub and Zenodo: https://github.com/AlanZambrano/Zambrano_et_al_2021.git.

AlanZambrano. (2022). AlanZambrano/Zambrano_et_al_2021: Zambrano_et_al_2021 (V.1). Zenodo. https://doi.org/10.5281/zenodo.7351594

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
