# Peer review of "A tragedy of the commons case study: modeling the fishers king crab system in Southern Chile"

_PeerJ, doi:10.7717/peerj.14906_

## Round 0.1 · original submission · Major Revisions

Dear authors,

My apologies for the prolonged length of the review process for your manuscript. After receiving comments from three reviewers, I believe this paper is potentially publishable in PeerJ following major revisions.

The main reviewer critique that needs to be addressed prior to publication is the lack of context for the paper. The discussion has no citations and the remainder of the paper is lightly sourced. This will likely require quite a bit of restructuring and additional writing in the discussion to tie your results to the big picture and to other research.

In addition, I believe a proficient English speaker should read through your revised manuscript prior to submitting your revisions. There are a number of grammatical issues that make the manuscript difficult to read.

When you resubmit your revisions, please submit a clean version of the manuscript, a tracked changes version of the manuscript, and an itemized reviewer response document.

Please let me know if you have any questions going forward.

Best,

Brandon P. Hedrick, Ph.D.

Reviewer 1 ·

Basic reporting

This manuscript presents some interesting analysis and findings with respect to a Game Theory simulation for the king crab fishery in Chile. However, this manuscript as it stands does not advance the literature on understanding fisheries, in large part because the research is not in any way framed in the context of existing literature on game theory, fishers' behavior and decision making, fisheries management, or fisheries biology. The authors have inserted several citations in the first couple paragraphs of the Introduction without any real context - for example, mentioning the Prisoner's Dilemma game and then including several references to papers that presumably include Prisoner's Dilemma games, but without any further discussion or inclusion of these references into the paper beyond an initial list. Likewise, the discussion and conclusions sections are entirely devoid of any references, suggesting the authors have not given any consideration to how their work falls into the scope of the work many others have done around this topic. There are many authors who have studied cooperation and competition among fishers, including using game theoretic approaches (see, for example, Hatcher et al. 2000; King and Sutinen, 2010; Raakjaer Nielsen, 2003; Sutinen and Kuperan, 1999; the work of Elinor Ostrom, and others).

The structure of this article is also insufficient for publication. The Discussion section is essentially a continuation of the Results section, describing the results in more detail without putting them into any sort of greater context as should be done in a discussion. The Conclusion also just repeats the results. This is where the authors should bring in some of the literature to frame their findings in a greater context. The Introduction is also clunky. I would specifically recommend bringing the section of the introduction describing the Fishery and the Fishers (and these two should be combined - no need to have them described separately) to below the third paragraph (line 62), to fully describe the fishery before then describing the use of game theory.

Finally, much of the language used in this paper is challenging to read, with many awkwardly constructed sentences and phrases, and many grammatical errors. The opening paragraphs of the Introduction, which should be drawing the reader in, are especially poor. For example, the first line of the manuscript reads "A common group of fisheries problems is widely recognized", which is poorly worded and ambiguous. The authors do not then describe what these fisheries problems are, so the reader is left to imagine what types of fisheries problems they may be referring to. I would suggest, if the authors are to attempt to resubmit this article, that they perhaps seek the assistance of an editor to help with phrasing and the English language.

Experimental design

The research and analysis seems to have been conducted sufficiently well, and the research design seems appropriate. The research is not adequately (and really not at all) tied in to previous research in this area, with no references provided to other existing models of fisher behavior and competition/cooperation, of which there are many. The research questions are well defined, but I'm not sure they are sufficiently answered, particularly the final question regarding regulations to ensure sustainability. While a full analysis of different fisheries management scenarios is undoubtedly beyond the scope of this manuscript, the authors do not demonstrate sufficient understanding of the types of regulations described in the paper and how fisher behavior may change in response to such regulations.

Validity of the findings

The findings of the analysis are well described and the conclusions link to the original research questions. However, again, the manuscript does not then couch the findings in the context of the literature, and so the validity is difficult to assess without an understanding of how these results fit in with other similar research.

This paper seems to ignore any considerations of how a fishery works outside of the context of a model. Outside of the game theory discussion, the authors consider only extending seasonal closures or implementing some form of shares (meaning quotas?) into the fishery. In reality, with a shorter season fishers will typically fish harder, so depending on other considerations such as the catchability of the species this may not significantly change the CPUE. The authors make some conclusions about the extraction rate that seem to assume that the extraction rate will be held constant, when in reality there are numerous external factors that determine the extraction rate. Competition with other fishers affects not only the catch and resource availability but also the price, which fluctuates based on demand but also on supply, depending on how many crabs are being caught. A longer season can conversely mean fishers catch the same amount but landings are more spread out and the ex-vessel price is higher. Additionally, the discussion of the Allee effect findings seems to ignore some basic biological principles of the king crabs. For example, they assume that the catch rate will go down for all fishers at the same rate if the population size is reduced. This is assuming the resource is spatially homogenous which it’s not likely to be. The resource will concentrate in areas of preferred habitat which may make exploitation easier in certain areas that others when population falls below a certain threshold. The authors may not be able to account for this spatial complexity in their model, but the paper should then describe some of this complexity and some of the caveats to their model that may affect their findings. All of this can easily be found in the literature, which, again, is not cited in this manuscript.

Reviewer 2 ·

Basic reporting

I am wondering if this type of modeling been done before for any other fishery? Or any other natural resource?
If yes, then it would be helpful to include a short summary paragraph of previous results.

I also suggest including a bit more background on the fishery. Why is this fishery important to study? Why is there so much illegal activity, is it because it’s a high value fishery? What level of enforcement to stop illegal fishing exists? Basically more background for the reader to better understand why it was worth carrying out this research.

If the stock is still abundant, then why are landings declining? More being caught illegally? Stock really isn't abundant?

Experimental design

More information is needed on how this research is fulfilling a knowledge gap.

The authors consider social pressure but not increasing enforcement (either at sea or factories). Why not? If unregistered boats are more likely to be caught then the attraction of acting as a super fisher declines. If illegal fishing is primarily motivated by money, then increasing the risks can act as a deterrent.

Validity of the findings

This is the major area of issue for me. As is currently written, the discussion does not contain any citations of other literature, nor does it link to other relevant research. I believe the findings are useful and the approach is novel. However a good discussion needs to go beyond the specific findings and put them in the context of other literature. What areas of research does this study advance? What questions does it help answer? How does it tie into the larger questions of fisheries compliance and sustainability?

The issue of short-term gain vs. long-term sustainability is one that applies to pretty much every area of natural resource management. People often cannot afford to place much importance on the long-term when they need the resources (including money) right away just to meet basic needs.

A specific note, the impact of social parameters section reads like results and should probably be moved into that section.

Additional comments

This work by the author presents an interesting approach to the problem of non-compliance in fisheries.

Reviewer 3 ·

Basic reporting

This manuscript employs a number of modelling approaches to highlight the potential impacts of different fisher behaviour, stock productivity levels and management controls (i.e. season length) on the sustainability of Chilean king crab.
The manuscript is generally well-written, but would benefit from input by a professional scientific editor, particularly to try and simplify the language wherever possible. For example, "season period fraction span" could be shortened to "season length".
The figures are generally clear and well labelled, although the light colours used in Figures 3 and 9 are very difficult to see and should be changed to darker colours.
My primary criticism of the paper is that it introduces some analyses and results in the Discussion section, specifically the text relating to Figure 12. I suggest the manuscript be re-structured such that this text and Figure 12 be moved to the Methods and Results sections.

Experimental design

The manuscript falls within the Aims and Scope of the journal. The research questions are reasonably well defined and the modelling approaches appropriate for the questions posed. The modelling is rigorous, but the explanation of the methodology needs to be clearer and simpler.

Validity of the findings

Much of the Conclusion section just reiterates the methodology. This section should be very brief and outline the key findings of the research and their implications for the management of the Chilean king crab fishery.

Additional comments

Nil

---

## Round 0.2 · Minor Revisions

Dear authors,

Thanks for your submission to PeerJ. The reviewers and myself find this to be a much improved version of your original submission following substantial revisions. There are still a few places that need to be revised to help shorten the paper and increase readability. However, following those revisions, I believe this paper will be publishable in PeerJ.

With your revision, please submit a tracked changes version of the manuscript, a clean version, as well as a response to reviewers document outlining the changes made.

In addition to the reviewer comments, I have added in a few comments below.

Expand the discussion of SSF in the introduction a bit. Why is this interesting and worth looking at in particular? I think this could be added after line 73 at the end of paragraph 2.

Line 64: ‘resource sustainability’

Lines 104–116: It seems like this should be a single paragraph.

Line 146–148: The centolla population is not mentioned at all until the end of the introduction. I think this should be integrated into the body paragraphs of the intro. Why choose centolla to ask this question? I think the first part of the methods answers this well, but a few sentences in the introduction would be valuable.

Line 650–655: I think it’s fine to discuss what your work adds, but not generally necessary to say how it is the first study to take a particular approach.

Line 693: What do you mean here by ‘not sufficient’? Not sufficient to maintain sustainability in the absence of reducing fishing season duration?

Line 926–930: Be sure to thank reviewers.


Figure 2 caption has grammar errors and is hard to follow. Please rewrite

I’m having a hard time following figure 4. Can you add in some text saying ‘move from super fisher to standard fisher’ or something on the plot itself? Having the Yes and No on the y-axis with the top bar makes it a bit hard to read. Some color might also help.


Please let me know if you have any questions.

Best,

Brandon P. Hedrick, Ph.D.

Reviewer 1 ·

Basic reporting

This version of the paper is significantly improved from the original manuscript in both form and content. The authors have added more background and contextualized the research and findings with existing literature references.
There are still a few areas where the language is unclear or could be written more clearly. The authors should continue to work with an editor to clean up the language.
The manuscript could still use a bit of restructuring. Primarily, there are a few areas, most notably in the introduction, where the text contains redundancies, or where the authors ask the reader to recall something that was mentioned in an earlier paragraph. These paragraphs could be restructured to avoid redundancies and perhaps slightly shorten the paper overall.
Some suggestions for cleaning up the text follow:
Line 104: beginning "As stated before" - this paragraph doesn't really provide new information, so should be combined with the earlier mention of this concept.
Line 114 discusses the Tragedy of the Commons as an example of the Prisoner's Dilemma- redundant with the 2nd paragraph of the Introduction which says the same thing.
Line 120: beginning "We assume that..." - this sentence refers to an assumption of the model and belongs in the Methods
Line 128: "While the Prisoner's Dilemma game..." This has already been discussed in the 4th paragraph, with different references. This is again redundant.
Lines 135-142: The content in this paragraph seems like a better fit for the Discussion
Line 151: The original manuscript referred to the species as Chilean king crab. It makes sense to refer to them as 'centolla' throughout, but elsewhere in the manuscript they are referred to as king crabs. Right up front this should say what kind of species centolla is, including that they are crabs.
Line 193, Interactions among fishers...: This is missing a description of WHY the authors are doing this study. E.g., In order to understand the decisions of super fishers, we set up a prisoner's dilemma game...
Line 358: The term "vessels" should probably be used throughout rather than boats, as this is more customary
Line 365: 'slow dynamics of the centolla' - the authors should clarify what they mean here. I assume this refers to population dynamics of the species?
Line 416 - In Short: Nothing about this Methods section is short, in fact it is quite long. I wonder if any elements of this section can be shortened. I recognize that this is really the heart of the paper, and the parameterization and assumptions of the model need to be discussed in detail. This last section is summarizing the Methods - really this should be summarized at the beginning of the section to guide the reader as to what will be described. Here it is just redundant.
Line 480 - Net extraction rate and Centolla dynamics: It would seem that the Results don't really begin until here - everything before this point probably belongs in the Methods section.

Experimental design

The experimental design is robust.

Validity of the findings

The findings are valid and robust; the conclusions link back to the findings.

Reviewer 2 ·

Basic reporting

Overall, the manuscript is much improved, and reviewer comments and suggestions incorporated. The introduction is much improved. There is clearer context in which this study is placed, and a separate section on the case study. The need for this study is better justified and the research questions are explicitly stated. The discussion has also been substantially improved. The revised manuscript now better places the study and its results in a broader context. I have a few comments, but they are minor.

Experimental design

No comment.

Validity of the findings

Lines 57-58: “However, ‘tragedy’ is not a universal feature of all fisheries for a number of reasons, including the fact that individual interests are often subservient to the collective interests of a community.” Good idea to provide citations here. Check the citations you already have as I’m sure a few of them would work.

Lines 71-72: “SSF are particularly exposed to the adverse effects of illegal fishing and noncompliance”. Why?

Lines 190-191: “it has exhibited decreasing landings despite the suspected increasing
fishing effort (rising number of traps per fishing boat)”. This suggests the stock is decreasing/overfished, which makes it a suitable fishery for a case study (i.e., concerns over sustainability).

Lines 256-257: “mainly because the unregistered fisher has no legal access to the resource
and still needs to work as a fisher.” This brings up an important point. What are the barriers to becoming a legal fisher, at the individual level (i.e., besides the cap on the number of boats)? Is the registration/fees expensive (compared to the average income)? Most people want to do the right thing and comply (as you have noted) so it might be useful to understand potential difficulties in having a registered boat.


Lines 739-740: “our results indeed suggest that reasonable king´s crab population levels can be maintained even in the presence of illegal fishers and super fishers.” Since the king crab stock is declining, what does that suggest about where the fishery currently sits on your model? Is the number of illegal fishers greater than the maximum allowed to sustain the population as mentioned above? I think your paper would be strengthened by exploring this, tying the model results to the known state of the fishery. Just a few sentences.

Additional comments

Pay attention when you are citing an author in the sentence. The author’s name should not be in (). This occurs several times. Example:

Line 654: “Ethical aspects of illegal fishing are discussed in (Satizábal et al., 2021)”. Change to: Ethical aspects of illegal fishing are discussed in Satizábal et al. (2021)

---

## Round 0.3 · Minor Revisions

Dear authors,

Thank you for your submission to PeerJ. I appreciate your careful attention to reviewer comments from the previous round of reviews. I now find this manuscript to be publishable in PeerJ pending just a few minor grammar changes below.

Paragraphs 2 and 3 are pretty repetitive. Halfway down in both paragraphs, you explain why you’ve chosen the crab fishery in particular. I think this should be tightened up to avoid repetition.

Line 205: should be ‘crab’ rather than ‘grab’

Line 221: What is ‘non-nule’?

Figure 3. What are the units for the y axis here?

Lines 664–666: Sentence is repeated by mistake

Line 765: ‘…there are a range of…’

Please let me know if you have any additional questions and I would be happy to answer them.

Best,

Brandon P. Hedrick, Ph.D.

---

## Round 0.4 · accepted · Accept

Dear authors,

Thank you for making these final revisions!

Best,

Brandon P. Hedrick, Ph.D.